# The Prevalence of Delirium in An Older Acute Surgical Population and Its Effect on Outcome

**DOI:** 10.3390/geriatrics4040057

**Published:** 2019-10-16

**Authors:** Jonathan Hewitt, Stephanie Owen, Ben R. Carter, Michael J. Stechman, Hui Sian Tay, Matthew Greig, Caroline McCormack, Lyndsay Pearce, Kathryn McCarthy, Phyo K. Myint, Susan J. Moug

**Affiliations:** 1Division of Population Medicine, Cardiff University, Heath Park, Cardiff CF14 4XW, Wales, UK; stephanie.owen2@wales.nhs.uk; 2Department of Biostatistics and Health Informatics, Kings College London, 16 De Crespigny Park, London SE5 8AF, UK; ben.carter@kcl.ac.uk; 3Department of Surgery, University Hospital of Wales, Heath Park, Cardiff CF14 4XW, Wales, UK; Michael.stechman@wales.nhs.uk; 4Department of Medicine for the Elderly, NHS Grampian, Aberdeen AB25 2ZN, UK; huisiantay@nhs.net (H.S.T.); matthew.greig@nhs.net (M.G.); caroline.mccormack@nhs.uk (C.M.); phyo.myint@abdn.ac.uk (P.K.M.); 5Department of Surgery, Salford Royal Infirmary, Stott Ln, Salford M6 8HD, UK; lpearce@doctors.org.uk; 6Department of Surgery, North Bristol NHS Trust, Bristol, Southmead Rd, Bristol BS10 5NB, UK; drkathrynmccarthy@hotmail.co.uk; 7Department of Surgery, Royal Alexandra Hospital, Paisley G12 9PF, UK; susanmoug@nhs.net

**Keywords:** Cognitive impairment, delirium, older surgical patients

## Abstract

Background: With an ageing population, an increasing number of older adults are admitted for assessment to acute surgical units. Older adults have specific factors that may influence outcomes, one of which is delirium (acute cognitive impairment). Objectives: To establish the prevalence of delirium on admission in an older acute surgical population and its effect on mortality. Secondary outcomes investigated include hospital readmission and length of hospital stay. Method: This observational multi-centre study investigated consecutive patients, ≥65 years, admitted to the acute surgical units of five UK hospitals during an eight-week period. On admission the Confusion Assessment Method (CAM) score was performed to detect delirium. The effect of delirium on important clinical outcomes was investigated using tests of association and logistic regression models. Results: The cohort consisted of 411 patients with a mean age of 77.3 years (SD 8.1). The prevalence of admission delirium was 8.8% (95% CI 6.2–11.9%) and cognitive impairment was 70.3% (95% CI 65.6–74.7%). The delirious group were not more likely to die at 30 or 90 days (OR 1.1, 95% CI 0.2 to 5.1, *p* = 0.67; OR 1.4, 95% CI 0.4 to 4.1. *p* = 0.82) or to be readmitted within 30 days of discharge (OR 0.9, 95% CI 0.4 to 2.2, *p* = 0.89). Length of hospital stay was significantly longer in the delirious group (median 8 vs. 5 days respectively, *p* = 0.009). Conclusion: Admission delirium occurs in just under 10% of older people admitted to acute surgical units, resulting in significantly longer hospital stays.

## 1. Introduction

Increased patient expectations in combination with anaesthetic and surgical advances are responsible for the increasing number of older patients being admitted to hospital for surgical management [1,2]. Older people (≥65 years) comprise a vulnerable group due to the multiple comorbidities they present with, and the role these play in determining peri-operative outcomes [3]. It is therefore important for clinicians to be aware of factors that adversely influence outcomes in this group. One such emerging risk factor is delirium [4], which is being increasingly recognised [5].

Delirium is acute cognitive impairment, differing from dementia (chronic cognitive impairment, which represents a broad spectrum of disease from mild to very severe) as it is a clinical syndrome characterised by a fluctuating course, disturbed consciousness and reduced ability to focus, sustain or shift attention, which is not explained by the presence of a pre-existing, established or developing neurological disorder [6,7]. Factors known to predict delirium in older people are multifactorial and include visual impairment, severe illness, pre-existing cognitive impairment, hypoglycaemia and a high blood urea nitrogen/creatinine ratio [8,9]. Delirium is common, affecting up to 50% of hospitalised patients ≥65 years [10]. In relation to delirium diagnosed on admission, a study in a medical population estimated 12% of patients were delirious, finding this to be associated with a composite end point of death and nursing home placement, but not death alone [11].

In recent years, delirium in surgical populations has been increasingly investigated, however much of this work focuses on post-operative delirium and its effect on outcome. Sprung et al. demonstrated that pre-existing cognitive impairment was a risk factor for post-operative delirium [12]. In elective colorectal surgery, other known risk factors for the development of post-operative delirium include increasing age, intraoperative hypotension, alcohol excess and blood loss [13]. Jones et al. also demonstrated in elective surgical patients that pre-existing cognitive impairment was highly predicted of post-operative delirium [14]. Furthermore, Vasunilashorn et al. found that delirium measures that consider both the intensity and duration of the delirium episode are the best predictors of poor outcomes in both surgical and medical patients [15]. 

With regards to older people undergoing elective abdominal surgery, interventions that involve orienting communication, nutrition and early mobilisation have proven effective in reducing post-operative delirium and reducing length of hospital stay [16]. Similarly, the use of a comprehensive geriatric assessment in patients undergoing elective vascular surgery proved effective in reducing length of stay [17]. However, Ansaloni et al. demonstrated that delirium incidence is significantly higher in emergency surgical patients compared to elective patients in the post-operative period (17.9% vs. 6.7%, *p* = 0.003) [18]. 

Delirium is becoming increasingly important in surgical decision-making [19], because approximately 40% of all delirium cases are estimated to be avoidable [20]. If an association between pre-operative delirium and poorer surgical outcomes is evident in an acute surgical population, this would necessitate the need to implement delirium screening into all older people surgical admissions in order to optimise patient management, safety and support the involvement of geriatricians in the acute management of surgical patients.

This study aimed to establish the prevalence of delirium on admission to hospital in older acute surgical patients, and whether admission delirium in this cohort is associated with 30- and 90-day mortality. Furthermore, the influence that admission delirium on length of hospital stay and readmission rates was analysed.

## 2. Materials and Methods 

As part of the Older Persons Surgical Outcomes Collaboration (OPSOC, http://www.opsoc.eu) this observational study was conducted across five UK hospitals [21]. Data were collected for consecutive patients ≥65 years, which presented to the acute general surgical unit of each study site throughout May and June 2014. In the UK, acute general surgery comprises all patients who are diagnosed as having a surgical (or potential surgical) diagnosis. The range of diagnoses is predominantly abdominal pathology and excludes urology, vascular, orthopaedic and neurological surgical disease. These patients remain under the management of a general surgical team, regardless of whether they ultimately undergo a surgical procedure. Data collection for the 2014 round pre-specified a focus on delirium. There were no elective surgical patients enrolled.

Baseline demographics were recorded. To assess comorbidities, we recorded; anaemia (haemoglobin < 129 g/L), hypoalbuminaemia (albumin < 35 g/L) and polypharmacy of (≥5 medications on admission). The included patient characteristics are presented by site.

Within 24 h of admission, and prior to any surgical intervention, participants received two consecutive cognitive function tests, the Confusion Assessment Method (CAM) and the Montreal Cognitive Assessment (MoCA). Unit staff (limited to medical students, nurses and doctors) at each hospital site were trained by the lead clinician for each site and assessed in performing the named cognitive tests prior to any data collection. The CAM is a simple to use, validated, four point score used to detect delirium (Appendix A) [22]. The MoCA is a validated 30-point questionnaire with a sensitivity of 90% for detecting mild cognitive impairment (Appendix A) [23]. Scores of 26 and above are considered normal. For participants unable to complete the MoCA the reason was recorded. 

Follow-up data were obtained from the in-hospital electronic clinical record: length of hospital stay (or time to death) was calculated as whole day integers, rounded up to the nearest day; survival status at Day 30 and Day 90 post-admission and hospital readmission within 30 days of discharge was noted. 

Data were recorded and stored in conjunction with local data management standard operating procedures. All participants were service users and both the MoCA and the CAM are freely available for routine use. As such, this study was deemed a service evaluation and did not require ethical approval and was registered at each participating site according to local guidelines.

Descriptive statistics were used to describe the cohort in terms of baseline characteristics and to establish the prevalence of cognitive impairment and delirium. Chi-squared tests (or Fisher’s exact tests) were used to assess whether the prevalence of baseline characteristics differed amongst: delirious and non-delirious groups; death at day 30; death at day 90; and readmission at day 30. The median length of hospital stay and age were compared using Mann–Whitney U tests. Univariable logistic regression models were used to assess delirium and outcome. Data analysis was conducted using Stata version 13.

To test the robustness of the non-parametric findings linking patients who were delirious to an increased the length of stay, an additional sensitivity analysis was carried out using a parametric log-transformed length of stay. The transformed length of stay was fitted using a general linear model, adjusted for age and sex. A multivariable logistic regression adjusting for patient age, sex, and MoCA score estimated an adjusted odds ratio (aOR) was additionally presented alongside the crude unadjusted OR.

## 3. Results 

Data were collected for 413 patients. Two patients were lost to follow up giving a sample size of 411 patients. In total, 146 (35.5%) patients were enrolled from the Aberdeen Royal Infirmary, 92 (22.4%) from the North Bristol NHS Trust, 77 (18.7%) from the University Hospital of Wales, Cardiff, 51 (12.4%) from the Central Manchester NHS Trust and 45 (10.9%) from the Royal Alexandra Hospital, Paisley (Appendix A).

The mean age of the cohort was 77.3 years old (SD 8.1). There were 212 (51.6%) women.

Of all 411 individuals analysed, 36 (8.8%; 95% CI 6.2–11.9%) were delirious and 367 (89.3%) were not delirious on admission. The remaining 8 (1.9%) individuals had missing CAM data. 

Overall, 356 (86.6%) people completed the MoCA. The scores ranged with a mean of 21.1 (median = 23, interquartile range of 18 to 26). The prevalence of cognitive impairment in this older acute surgical population was 70.3% (95% CI 65.6–74.7%). Of the 55 individuals unable to complete the MoCA, 15 (27.3%) had known dementia, 6 (10.9%) had visual impairment, 3 (5.5%) had hearing impairment, 2 (3.6%) had physical impairment, 26 (47.3%) had another reason for which they could not complete the MoCA, and 3 (5.5%) had no reason recorded (see Table 1). The reasons for having missing CAM data were not recorded.

In terms of baseline characteristics, 183 (44.5%) patients were anaemic, 171 (41.6%) had hypoalbuminaemia, and 279 (67.9%) were taking more than five medications on admission. In total, 79 (19.2%) individuals underwent surgery during their admission. The baseline characteristics of the cohort with regards to the exposure variable (delirium) are given in Table 2. The delirious group were significantly older than the non-delirious group (median ages were 81.6 years vs. 76.9 years, *p* < 0.001). 

In total, 20 (4.9%) patients died within 30 days of admission, of which 2 (10%) were delirious and 18 (90%) were not delirious. One patient had missing data for death at 90 days. Of the remaining 410 individuals, 36 (8.8%) died, of which 4 (11.1%) were delirious, 31 (86.1%) were not delirious, and 1 (2.8%) had missing CAM data. Readmission rates were recorded for all 411 patients, and 82 (20.0%) were readmitted within 30 days of discharge. The OR for these outcomes, based on the presence of delirium are shown in Table 3. The associations between all baseline characteristics and clinical outcomes are presented in Table 2.

Age was associated with 30-day mortality (*p* = 0.03) and differed significantly amongst the delirious and non-delirious groups (*p* < 0.001). 

A total of 11 patients had missing data for their length of hospital stay. For the remaining 400 patients, the length of stay ranged between 1–90 days (Appendix A) with a median of 5.5 days. There was a significant difference between the delirious (8 days) and non-delirious (5 days) groups (*p* = 0.009).

### Sensitivity Analysis

The parametric analysis of the log-transformed length of stay found that those who were delirious experienced a longer length of stay *(p* = 0.02), after being adjusted by age and sex.

## 4. Discussion

The prevalence of delirium on admission in this acute surgical population was 8.8% and was associated with an increased length of stay but not mortality or readmission to hospital. As expected, patients with delirium were older than those without (*p* < 0.001) [24,25]. There were no significant differences evident between the delirious and non-delirious groups for any of the other baseline characteristics measured. The proportion of delirious and non-delirious individuals receiving surgery did not differ significantly. Additionally, despite a high prevalence of cognitive impairment in the acute general surgical population (70.3%), the prevalence of delirium was low, present in less than 10% of all older patients. The comparatively high prevalence of cognitive impairment is likely to reflect the tool used to detect it; namely the MoCA. This test is specifically sensitive in the detection of mild cognitive impairment and therefore would be expected to pick up a high number of cases.

This is the first study to assess the prevalence of delirium on admission in an older adult acute surgical population. Previous studies have reported on delirium development during hospital stay (incident delirium) [26], its development post-operatively or in orthopaedic surgery [27,28,29]. In a pre-operative database analysis, by Gajdos et al. [30], the significance of pre-operative impaired sensorium (IS, an alteration in individual’s consciousness or mental awareness) on surgical outcomes in an elective general surgical population was investigated and reported. The prevalence of IS was 0.6% with a significantly higher mortality rate in those patients with IS (14.9% vs. 20.4%, *p* < 0.001). Our prevalence rate was much higher (8.8%). This is likely to reflect the active detection delirium, which has a wider definition than IS, in our older emergency population, combined with a likely underreporting of IS within the database.

Our study did not demonstrate any association between admission delirium and early or medium term mortality. Other studies have investigated the longer-term prognosis of older hospitalised patients with admission delirium. George et al. report that at one year, patients with medical conditions and admission delirium had increased mortality (OR 12.62, 95% CI 2.86–114.04, *p* < 0.001) and were more likely to be readmitted to hospital (OR 2.05, 95% CI 1.19–3.54, *p* = 0.008) [30]. Further, in the acute medical settings, Inouye et al. stated that “delirium is an important independent prognostic determinant of hospital outcomes”. Their study demonstrated an association with a composite endpoint of death and nursing home placement [31]. A weakness of our study and limitation that needs to be highlighted is that study was not powered to detect any differences and the wide confidence intervals generated highlight the imprecision of the estimates. Further, additional variables, for example, poor vision or hearing, can contribute to the development of delirium. We did not collect these data and therefore it is likely that some additional factors may have contributed to the development of delirium. Future studies should try to collect data on as many potentially contributing variables as is practical.

Ansaloni et al. [18] analysed post-operative delirium in a general surgical population. Although they did not analyse pre-operative delirium in their mixed population of 351 elective and emergency general surgical patients, they found an incidence of 13.2% of post-operative delirium. Significantly, delirium was higher in the group undergoing emergency surgery (17.9%). This study analysed their elective and emergency surgical patients together so separate conclusions about the emergency population alone are difficult to generalise, however their study showed an increase in mortality for people with post-operative delirium. 

Our other outcome of interest was length of hospital stay, which was found to be increased by three days in people with admission delirium. This is the first study to assess this outcome variable. The Ansaloni study [18] discussed in the previous paragraph provides the nearest comparable study; in their population of emergency and elective surgical patients, people who developed post-operative delirium stayed in hospital for 13 days longer than those who did not. 

Delirium is an easily identifiable, treatable condition, and so assessing for its presence during hospital admission could be useful in ensuring prompt diagnoses and treatment [8]. Additionally, delirium alone, or in addition to cognitive impairment, may hinder one’s ability to provide informed consent [27]. This is important in surgical settings, where surgical intervention may be required, as we need to know whether patients are in a position to provide consent for such procedures. Furthermore, assessing for delirium on admission to the acute surgical unit provides a baseline recording, making it easier to identify the development of both incident and post-operative delirium. [32] There is therefore a strong theoretical rational to screen for admission delirium, regardless of its effect on mortality. 

Further, future research identifying people with delirium diagnosed on admission to hospital should assess their immediate delirium status in the post-operative period. It is likely that these individuals make a large proportion of those diagnosed with post-operative delirium. If proven, this would identify a group of people where early intervention to treat delirium and ameliorate the detrimental effects of delirium could be targeted.

A limitation of this study is the failure to consider the past cognitive history of individuals found to be delirious. We have assumed the cognitive impairment detected is due to delirium, however it is possible that delirium can occur in addition to a pre-existing diagnosis of dementia. In order to differentiate amongst these varying aspects, future research should use the Informant Questionnaire on Cognitive Decline in the Elderly (IQCODE), a 26-point questionnaire that is used to differentiate between delirium and dementia [33]. Further, we used a single test for the assessment of delirium, the CAM score. While this is a valid and recognised test, other screening tools, such as the 4AT score, are available and may be more useful [34]. Also, while we performed CAM training in this study, this occurred at site level and was not centralised. This may have influenced the accuracy of the data. Additionally, the study did not collect data on the type of presenting illness that these surgical patients had. Therefore it is not possible to assess whether the severity of the presenting illness may have played in the presence of admission delirium. The study did not collect data on pre-existing conditions, such as cognitive impairment which is likely to have been a factor in the pre-admission development of delirium [14]. Few patients experienced mortality, and readmission, and these analyses were underpowered so it should be remembered that a lack of evidence is not evidence of no effect.

This multi-centre observational study describes the first attempt of establishing the prevalence of admission delirium and its influence on mortality with regards to an older acute general surgical population. Delirium was present on admission in 8.8% of patients, and led to a significantly longer hospital stay in this population. 

## Figures and Tables

**Table 1 geriatrics-04-00057-t001:** The prevalence of delirium and cognitive impairment.

CAM 411 (100%)	MoCA 411 (100%)
Delirious 36 (8.8)	Fail = score ≤ 25 253 (70.3)
Not delirious 367 (89.3)	Pass = score ≥ 26 103 (16.3)
Missing 8 (1.9)	Incomplete 55 (13.4)

**Table 2 geriatrics-04-00057-t002:** Included patient characteristics of the delirious and non-delirious group.

Included Patient Characteristics		Non-Delirious Group (CAM Negative) *N* = 367 (%)	Delirious Group (CAM Positive) *N* = 36 (%)	Total *N* = 403		Test Statistic
Age	Mean (S.D)	76.9 (8.0)	81.6 (7.9)	77.3 (8.1)		*p* = 0.001
Sex	Male	175 (47.7)	19 (52.8)	194	403	*p* = 0.683
	Female	192 (52.3)	17 (47.2)	209		
Polypharmacy (≥5 medications)	Yes	246 (67.0)	27 (75.0)	273	403	*p* = 0.329
	No	121 (33.0)	9 (25.0)	130		
Anaemia (Hb < 129 g/L)	Yes	161 (43.9)	18 (50.0)	179	403	*p* = 0.596
	No	206 (56.1)	18 (50.0)	224		
Hypoalbuminemia (Albumin < 35 g/L)	Yes	153 (41.7)	13 (36.1)	166	403	*p* = 0.637
	No	214 (58.3)	23 (63.9)	237		
Operation	Yes	70 (19.1)	7 (19.4)	77	403	*p* = 0.937
	No	290 (79.0)	28 (77.8)	318		
	Missing	7 (1.9)	1 (2.8)	8		

**Table 3 geriatrics-04-00057-t003:** Odds ratios to measure the association between delirium and outcomes.

Cohort Outcomes		Delirious Group (CAM Positive) *N* = 36 (%)	Non-Delirious Group (CAM Negative) *N* = 367 (%)	Total *N* = 403		Unadjusted OR (95% CI)	Adjusted OR * (95% CI)
Death at 30 days	Dead	2 (5.6)	18 (4.9)	20	403	1.14 (0.25–5.13) *p* = 0.70	0.80 (0.23–2.83) *p* = 0.73
	Alive	34 (94.4)	349 (95.1)	383			
Death at 90 days	Dead	4 (11.1)	31 (8.5)	35	402	1.35 (0.45–4.07) *p* = 0.82	0.41 (0.07–2.59) *p* = 0.34
	Alive	32 (88.9)	335 (91.5)	367			
Readmission 30 days after discharge	Yes	7 (19.4)	75 (20.4)	82	403	0.94 (0.40–2.23) *p* = 0.89	0.59 (0.17–2.11) *p* = 0.42
	No	29 (80.6)	292 (79.6)	321			

* Odds ratio adjusted for: age; sex; and MoCA.

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
