# Peer review of "The Prevalence of Delirium in An Older Acute Surgical Population and Its Effect on Outcome"

_geriatrics, 2019, doi:10.3390/geriatrics4040057_

Round 1

Reviewer 1 Report

1) The delirium prevalence on admission seems low given the type of population included. It is true that this is the first study to report delirium prevalence on admission. I would suggest to include additional information on the characteristics of patients included (e.g type of surgery, signs of infections on admission, WBC etc) to understand why the delirium rate is low compared to other population and other studies. For instance it could be that there is a low rate of people with dementia. The MOCA could indeed detect mild cognitive impairment. Indeed the mean MOCA score is quite high. Can you please provide also the median and IQR along with minimum and maximum score?

2)  What was the level of CAM training? It is well accepted that the delirium rate could be low if a formal CAM training is not performed before delirium assessment. I can see that different people actually performed the CAM. Do you have any pre and post-training data on the reliability of CAM assessment?

3) It is unclear if you actually assess every patients with the MOCA also if they were delirious. Please clarify and provide the numbers of cognitively impaired patients without delirium. It is also unclear why a person with dementia would be unable to perform the MOCA.

Author Response

Dear Editor

Many thanks for allowing us to respond to the comments, the document has been improved with the reviewers suggestions.

I attach a point by point response document, which covers the issues raised by both reviewers and an updated manuscript where the changes are highlighted using track changes

Regards

Jonathan Hewitt

Reviewer 2 Report

The authors conducted an observational study examining the prevalence of delirium upon admission to the hospital for a surgical intervention and investigated the association between prevalent delirium and 30- and 90-day mortality, readmission, and length of stay. This is an important topic as delirium is a dangerous hospital complication. The study and manuscript are overall straight forward. The manuscript would be improved if the authors addressed the following: 

Introduction

The overall introduction is difficult to follow and it seems that there may be missing words/linking sentences. The manuscript would be greatly improved with up-to-date references. Much work has been published in perioperative delirium in the last 5 years. For example, Jones et al (2016) demonstrated that pre-existing cognitive impairment was strongly associated with the risk of postoperative delirium - this should be cited along with Sprung et al. This is similar to the reference to nonpharmacological delirium prevention measures - Hshiesh et al (2015) conducted a meta-analysis on such interventions as well as Siddiqi et al (2016) and included all surgery types. Siddiqi also separates out nonpharm effectiveness in emergency surgery (typically hip fracture) vs elective surgery. 

The authors should incorporate the DSM 5 definition of delirium

The authors switch between elective and emergent surgery as well as medical populations in reference to delirium. It would be easier to follow and understand if the authors stuck to their population of interest, emergency surgery in older adults. 

Methods

What were the inclusion/exclusion criteria? Was informed consent performed? How were patients with dementia handled? Proxy consent? 

Is there a reason why the type of surgery, length to surgery, a comorbidity index, and an overall assessment of illness severity was not included? Including the type of surgery would improve generalizability of study results.

Regarding the statistical analysis, is there a reason the authors did not use Cox regression or multivariate logistical regression to investigate the associations between delirium and mortality? This would allow the inclusion of age, sex, MoCA score, and type of surgery. The current analysis is rudimentary and the manuscript/results would be more robust with the inclusion of covariates in modeling procedures. 

Was a power analysis done beforehand? 

Results

In Table 2, it appears that ~79% of patients did not undergo surgery. This is confusing as I thought that patients were admitted to an acute surgical unit and the intent of the study was to determine the prevalence of delirium on admission to the hospital in older acute surgical patients. However, if only ~20% had surgery - why were the others admitted and included in the study? Are the others medical patients? This might explain the low prevalence of admission delirium in this study. 

Table 2 - univariate analysis examining the differences in MoCA score between the delirious/nondelirious cohort should be included. Surgery type should also be included if it was collected. 

Table 3. The authors show an OR that is adjusted for age. How was this done? This needs to be explained in the statistical methods section. 

Discussion

The authors write that age was an expected finding. Age was not mentioned in the introduction as a risk factor and if it was an expected finding, it should be mentioned with the list of other risk factors in the introduction. 

The authors mention that there were "no significant differences evident between the delirious and non-delirious groups for any of the other baseline characteristics measured." however I do not see a univariate test for the MoCA between delirium/non-delirious. This is very important to include and discuss. Was pre-existing cognitive impairment associated with admission delirium? 

This is not the first study to report on the prevalence of delirium on admission in an older adult acute surgical population. Freter et al (2005 & 2015) reported on prevalent delirium prior to hip fracture surgery, Kalisvaart (2006) examined both emergent hip fracture surgery and hip replacement surgeries, Moerman (2012) reported on hip fracture surgery and pre/post delirium, Muangpaisan et al (2015), Furlaneto & Garcez-Leme (2006) also reported on the phenomenon. "Delirium in elderly people" by Sharon Inouye also reports on the prevalence of delirium in the orthopedic population. The authors may be one of the first to focus specifically on the prevalence in a broad surgical population, however, this is not highlighted as the surgery types are not reported. It is not known whether this population is largely hip fracture patients or a different type of emergent surgery. The authors need to revise the 2nd paragraph of the discussion section to reflect that other studies have reported on prevalent delirium in the perioperative population. 

Is "impaired sensorium" the same as "subsyndromal delirium"? It may be a better comparison to compare the delirium prevalence rate of this study with other studies that have reported on prevalent delirium in hip fracture patients - see list above for references. 

This is also not the first study to assess length of hospital stay. That is a well known outcome of delirium in general. The authors need to tone down their language and reference the bodies of work that has demonstrated an association between delirium (prevalent and incident) and hospital length of stay. 

Round 2

Reviewer 1 Report

The authors are correct in reporting that the absence of data on delirium incidence but the prevalence is relatively low. I would suspect that the low prevalence might be linked to the actual CAM training and not clear agreement on delirium assessment. I would add this information. 

Author Response

We note this and have added a line to the end as a limitation in the discussion

Reviewer 2 Report

Thank you to the authors for considering my comments and feedback. I think the manuscript is much improved from the original version. Minor text editing is needed throughout the document. Please review reporting guidelines on statistical p-values and odds ratios in the AMA manual of style. 

Author Response

We note this important point, we will be thoroughly guided my the copy editors as to the style and presentation they wish the article to take.